

# Bacterial and archaeal spatial distribution and its environmental drivers in an extremely haloalkaline soil at the landscape scale

Martha Adriana Martínez-Olivas[1], Norma G. Jiménez-Bueno[1], Juan Alfredo Hernández-García[2], Carmine Fusaro[3], Marco Luna-Guido[1], Yendi E. Navarro-Noya[4] and Luc Dendooven[1]

[1] Laboratory of Soil Ecology, Cinvestav, Mexico City, Mexico
[2] Laboratory of Biological Variation and Evolution, Department of Zoology, Escuela Nacional de Ciencias Biológicas, Instituto Politecnico Nacional, Mexico City, Mexico
[3] Centro Tlaxcala de Biología de la Conducta, Universidad Autónoma de Tlaxcala, Tlaxcala, Tlaxcala, Mexico
[4] Cátedras Conacyt, Universidad Autónoma de Tlaxcala, Tlaxcala, Tlaxcala, Mexico

Corresponding author
Luc Dendooven,
dendooven@me.com

## ABSTRACT

**Background:** A great number of studies have shown that the distribution of microorganisms in the soil is not random, but that their abundance changes along environmental gradients (spatial patterns). The present study examined the spatial variability of the physicochemical characteristics of an extreme alkaline saline soil and how they controlled the archaeal and bacterial communities so as to determine the main spatial community drivers.

**Methods:** The archaeal and bacterial community structure, and soil characteristics were determined at 13 points along a 211 m transect in the former lake Texcoco. Geostatistical techniques were used to describe spatial patterns of the microbial community and soil characteristics and determine soil properties that defined the prokaryotic community structure.

**Results:** A high variability in electrolytic conductivity (EC) and water content (WC) was found. Euryarchaeota dominated Archaea, except when the EC was low. Proteobacteria, Bacteroidetes and Actinobacteria were the dominant bacterial phyla independent of large variations in certain soil characteristics. Multivariate analysis showed that soil WC affected the archaeal community structure and a geostatistical analysis found that variation in the relative abundance of Euryarchaeota was controlled by EC. The bacterial alpha diversity was less controlled by soil characteristics at the scale of this study than the archaeal alpha diversity.

**Discussion:** Results indicated that WC and EC played a major role in driving the microbial communities distribution and scale and sampling strategies were important to define spatial patterns.

## INTRODUCTION

Soil is spatially and temporally the most heterogeneous environment on earth. It provides adequate surroundings for a large variety of microorganisms, whose interactions with environmental factors (detritus sphere, pH, hydrosphere, etc.) make soil an extremely complex reactor. At a global scale, the weather forms the soil, and as the scale decreases, the soil structure defines the microbial microenvironment diversity (*Lavalle & Spain, 2003*; *Martiny et al., 2006*; *Legendre & Legendre, 2012*).

Several studies have shown that in soil the distribution of microorganisms is not random, but their abundance and activity changes with environmental gradients forming spatial patterns. The study of microbial spatial patterns provides important information that can help to (1) predict the spatial distribution of microorganisms, (2) understand the diversity moulding and evolutionary processes, (3) understand the impact of human activities on biodiversity at different scales, (4) lead to the development of strategies for ecosystem management and (5) determine how microorganisms distributed in a specific area contribute to biogeochemical processes (*Fierer, 2008*; *Philippot et al., 2009*; *Bru et al., 2011*; *Fakruddin & Mannan, 2013*; *Rousk & Bengtson, 2014*). The use of omic techniques together with the application of geostatistical tools allows for the determination of the distribution patterns of microorganisms at different scales and their relationship with environmental or physicochemical factors (*Franklin & Mills, 2003*; *Philippot et al., 2009*). A high correlation between microbial communities and soil pH, electrolytic conductivity (EC) and salinity, have been found in several studies at scales that vary from a few cm's to km's in a wide range of soil types (*Andrew et al., 2012*; *Inskeep et al., 2013*; *Crits-Christoph et al., 2013*).

A large number of novel members of the Archaea Domain have been detected in extreme saline environments (*Oren, 2003*, *2016*; *Grant et al., 2011*). Halophilic Archaea of the Halobacteriaceae family are often the most dominant halophilic organisms in extreme saline environments (*Oren, 2003*; *Grant et al., 2011*). Most of its members can grow in at least 100–150 g salts l$^{-1}$. The hypersaline environments studied are mostly aqueous, such as solar salterns, dead sea and alkaline lakes in Egypt (*Oren, 2016*), but only a few have focused on soils or sediments. In some of these studies, Archaea diversity and richness increased with salinity or EC and correlated with pH, organic C and particle size, and Euryarchaeota were the dominant archaeal phylum (*Hollister et al., 2010*; *Canfora et al., 2015*). Weather conditions can alter soil salinity, and hence time and spatial variability are determinant factors (*Budakoglu et al., 2014*; *Canfora et al., 2015*).

The drained soil of the former lake Texcoco has characteristics that make it a unique and heterogeneous environment (*Dendooven et al., 2010*). In previous studies with soil of the former lake Texcoco, a great variability in pH, that is, 8.5–10.5 (*Navarro-Noya et al., 2015*), 7.8–10.0 (*Valenzuela-Encinas et al., 2009*) and 7.8 (*Valenzuela-Encinas et al., 2012*) and EC (0.7–157.2 *Navarro-Noya et al., 2015*; 0.68–159 *Valenzuela-Encinas et al., 2009* and 0.68 *Valenzuela-Encinas et al., 2012*) was found. *Valenzuela-Encinas et al. (2008)* and *Navarro-Noya et al. (2015)* studied the archaeal and bacterial diversity and phylogeny in soil of the former lake Texcoco. They found that Euryarchaeota dominated the archaeal

population and Proteobacteria the bacterial population. *Valenzuela-Encinas et al. (2009)* found that the dominant bacterial groups in this area belonged to the Alpha- and Gammaproteobacteria (mostly members of *Halomonas*), Bacteroidetes and Cyanobacteria. They also found that bacterial diversity and richness were greater in the soil of the former lake of Texcoco with 56 dS m$^{-1}$ EC compared with those of soil with 159 dS m$^{-1}$ EC values. No study, however, described how spatial variability would alter the microbial diversity in the former lake bed and if similar groups dominated along a gradient.

The objective of this work was to study the spatial distribution of the microbial communities in the soil of the former Lake Texcoco with a high pH and a high EC and to determine the soil characteristics that controlled their distribution. We hypothesized that in a 200 m transect in the soil of the former lake Texcoco, the soil microbial communities would be controlled by the variation in pH and EC.

## MATERIALS AND METHODS

### Site description and soil sampling

The sampling area in the former lake bed was defined where a high pH and EC were found previously (19.30°N, 98.53°W). Soil was collected on August 22nd, 2014. The temperature was 19 °C during sampling shortly after a period of precipitation, that is, 189.2 mm, according to the national meteorological system (Mexico) (http://smn.cna.gob.mx/es/).

The salt content of the former lake is the result of upwelling brackish groundwater, preferential water infiltration, evaporation and salt leaching, especially during the rainy season. Some parts of the former Texcoco lake bed have been drained, which contributed to variations in salt content (*Dendooven et al., 2010*). *Distichlis spicata* L., a salt resistant grass, can be found in patches in the former lake bed. It forms extensions from its base over the soil when the salt content is too high. Its decomposition will be hampered in those parts where the salt content is too high.

Soil was sampled from 13 points along a 211 m transect. A GPS unit (eTrex Vista® C Garmin, Olathe, KS, USA) was used to determine spacing between the sampling points. The one cm topsoil was removed and approximately 500 g soil samples were collected from the 1 to 15 cm layer. The samples were two mm sieved. A 20 g soil sub-sample was used to extract DNA and the rest was dried at room temperature for 24 h. The air-dried soil was analyzed for pH, water content (WC), EC, particle size distribution, total carbon and inorganic carbon.

Details and references to the methods used can be found in *Franco-Hernández et al. (2010)*. Soil pH was measured in 1:2.5 soil-H2O suspension using a glass electrode, the EC was determined in a saturated soil-paste extract (*Rhoades et al., 1989*) and total C was determined by oxidation with potassium dichromate ($K_2Cr_2O_7$) (*Amato, 1983*). The inorganic C was determined by adding five ml HCl 5M to one g soil and trapping the emitted $CO_2$ in a five ml NaOH as described by *Bundy & Bremner (1972)*. Soil particle size distribution was determined by the Bouyoucos method (*Gee & Bauder, 1986*). The WC was calculated from the weight loss after drying the samples in an oven at 105 °C for 24 h.

## DNA extraction, PCR amplification, and sequencing

The 20 g soil sub-samples were incubated separately in aerobic conditions for 7 days in a one l glass jars containing a vessel with 20 ml 1M NaOH to trap the $CO_2$ evolved and a vessel with 20 ml water to avoid water loss. This incubation was applied to minimize a possible effect of soil sampling on the microbial community structure. After a week, six sub-samples of 0.5 g soil of each incubated sample were taken and washed with 0.15M sodium pyrophosphate and 0.15M phosphate buffer (pH 8.0) to remove fulvic and humic acids (*Ceja-Navarro et al., 2010*). Two 0.5 g sub-samples were extracted for DNA with the technique described by *Valenzuela-Encinas et al. (2008)*, two with the method of *Sambrook & Russell (2001)* and two with the technique of *Hoffman & Winston (1987)*. As such, three g soil from each sampling point was extracted for DNA, that is, three extraction techniques applied in duplicate to 0.5 g soil. The duplicated extraction products from the three techniques were pooled and stored at −20 °C. A combination of several soil sampling and DNA extraction strategies was used to assure the whole metagenome was extracted.

Primers used to amplify the bacterial 16S rRNA V1-V6 gene region was targeted (approximately 900 bp) using 10-bp barcoded primers 8-F (5′-AGA GTT TGA TCI TGG CTC A-3′) and 949-R (5′-CCG TCW ATT KCT TTG AGT T-3′), and containing the A and B 454 FLX adapters (*Navarro-Noya et al., 2013*). The amplification program included 25 cycles of 45 s denaturation at 94 °C, alignment at 50.2 °C for 45 s and extension at 72 °C for 1 min 30 s. Equal amounts of the product of five reactions were pooled and purified using the DNA Clean & Concentrator purification kit as recommended by the manufacturer (Zymo Research, Irvine, CA, USA), and quantified using the PicoGreen® dsDNA assay (Invitrogen, Carlsbad, CA, USA) and the NanoDrop® 3300 Fluorospectrometer (Thermo Scientific NanoDrop; Thermo Fisher Scientific, Wilmington, DE, USA). The archaeal V1–V3 region was targeted (approximately 560 bp), using 10-bp barcoded primers A-25-F (5′-CYG GTT GAT CCT GCC RG-3′) and A-571-R (5′-GCT ACG GNY SCT TTA RGC-3′), and containing the corresponding A and B 454 FLX adapters. The amplification program contained 30 cycles and the amplification products were treated as those of Bacteria. Purification protocols and the quantification of the DNA are given in *Navarro-Noya et al. (2013)*. Sequencing was done by Macrogen Inc. (DNA Sequencing Service, Seoul, South Korea) using a Roche 454 GS-FLX Plus System pyrosequencer (Roche, Mannheim, Germany).

## Microbial community analyses

The analysis of the pyrosequencing data was done with the QIIME version 1.8.0 software pipeline (*Caporaso et al., 2010b*). First, poor and low quality sequences were removed from the data set. All sequences of Archaea with length <290 and >530 nt, quality score (Phred) <25, containing homopolymers >6, and with two minimal errors in barcodes and primers and those of Bacteria with length <380 and >930 nt, quality score (Phred) <25, containing homopolymers >6 and with two minimal errors in barcodes and primers, were removed.

The operational taxonomic units (OTUs) were defined at a 97% similarity level ($OTU_{97}$) using the Uclust algorithm (*Edgar, 2010*) and determined against the greengenes v13_5 database (http://greengenes.lbl.gov/). One representative sequence of each $OTU_{97}$ was selected, that is, rep-set. Chimeric sequences were detected using the chimera slayer and removed from the rep-set (*Haas et al., 2011*). Sequence alignments of the rep-set were done against the Greengenes core set using PyNAST and filtered at a 75% threshold (*Caporaso et al., 2010a*).

One representative sequence of each $OTU_{97}$ was chosen and the taxonomic assignment was determined using the naïve Bayesian rRNA classifier from the Ribosomal data project (http://rdp.cme.msu.edu/classifier/classifier.jsp) (*Wang et al., 2007*) with a confidence threshold of 80% (removing singletons and doubleton's) and based on the Greengenes v13_5 reference database (*DeSantis et al., 2006*). Biological observations matrices (biom, Cambridge, MA, USA) with the taxonomic assignments and metadata of the soil samples were constructed and used in further analysis.

The generated biom table was used to calculate the distributions at different taxonomic levels, while diversity and richness estimators were determined using data filtered and rarefied (*Kuczynski et al., 2011*). Richness estimators (Chao1, abundance-based coverage estimator (ACE)), alpha diversity (Simpson, Simpson E and Shannon), phylogenetic diversity (PD) and Good's coverage of counts for both Archaea and Bacteria in each sample, were calculated using the QIIME 1.8 pipeline, with the alpha_diversity.py script (*Chao, 1984*; *Chazdon et al., 1998*; *Chao, Hwang & Yang, 2000*; *Faith & Baker, 2007*). The biom matrices were rarefied to the minimum sequence found along the transect, 139 sequences for Archaea (Average: 732.462) and 125 sequences for Bacteria (Average: 302.167), to normalize the sequence number for each sample creating a subsampled $OTU_{97}$ table using the NumPy algorithm for pseudo-random sampling. Samples with fewer sequences than the requested rarefaction depth were omitted in the analysis.

The relative abundances were calculated for $OTU_{97}$ and genus taxonomic level in each sample (*Legendre & Legendre, 2012*).

## Statistical and geostatistical analyses

Descriptive statistics were determined for each physicochemical parameter in the transect using the R environment (*R Development Core Team, 2014*). The following statistics were applied: minimum, maximum, quartiles, standard deviation, variance and normality test (Kolmogorov–Smirnoff, Shapiro–Wilks and Q–Q plots). The kurtosis and skewness were calculated, and boxplot and tendency graphs were made for each factor.

Geographic exploratory analysis was done using GeoR, gstat and sp packages in the R environment to determine the spatial distribution of each physicochemical factor (values at every sampled point) (*Pebesma, 2004*; *Bivand & Pebesma, 2005*; *Ribeiro & Diglee, 2016*) The Mantel autocorrelograms were determined with the ncf package (*Bjornstad, 2016*), using 1,000 permutations. The omnidirectional semivariograms Eq. (1), were calculated using the geoR and gstat package within the R environment (*Pebesma, 2004*; *Bivand & Pebesma, 2005*). A cloud semivariogram to identify tendencies or atypical values, and a grouped semivariogram were computed for fitting. The semivariograms model

fit, which seeks to characterize them, describing how the parameters varied with the distance, was done using the Gstat library in the R environment.

$$\gamma(h) = \frac{1}{2N(h)} \sum_{\alpha=1}^{N(h)} [Z(u_\alpha) - Z(u_\alpha + h)]^2. \tag{1}$$

where $\gamma(h)$ was the semivariance, $h$ distance between the sample points, $Z$ the value of physicochemical factor, $N(h)$ the total pair of locations separated by a lag distance $h$, $Z(u_\alpha)$, and $Z(u_\alpha + h)$ values of $Z$ at positions $u_\alpha$ and $u_\alpha + h$ (*Goovaerts, 1998*).

Multiple correlations were done using the ncf package to select factors and to determine the correlations between them (*Bjornstad, 2016*). The interpolation method used was a standard form of kriging called ordinary kriging algorithm (*Isaaks & Srivastava, 1989*; *Goovaerts, 1998*) in which predictions are made Eq. (2):

$$\hat{Z}_{OK}(S_0) = \sum_{i=1}^{n} w_i(S_0) * z(S_i) = \lambda_0^T * z \tag{2}$$

where $\lambda_0$ is the vector of kriging weights ($w_i$) and $z$ is the vector of $n$ observations at primary locations (*Hengl, 2009*). The kriging was computed for a polygon of 0.71 Ha (7,141 m$^2$) composed of 3,000 pixels each of 2.38 m$^2$ around the sampling transect with the model of adjusted semivariograms for selected physicochemical factors using gstat package (*Pebesma, 2004*). Validation was done by leave-one out cross validation kriging in gstat (*Pebesma, 2004*; *Hengl, 2009*). The cross validation predicts the value at one location by kriging without using the observed value and doing it for all the points in the dataset one at a time so each point is assessed versus the whole (*Hengl, 2009*). Average standard error (ASE Eq. (3)) and root mean square error (RMSE, Eq. (4)) were calculated by a cross validation of predicted versus observed values. The kriging interpolation was considered good when the measures ASE and RMSE assessing variability of predicted values were similar. The normalized RMSE (RMSE$_r$, Eq. (5)) described the consistency of the model, explaining the variation of specific parameters. A value RMSE$_r$ close to 40% indicated an accurate prediction (*Hengl, 2009*; *Chabala, Mulolwa & Lungu, 2017*).

$$\text{ASE} = \sqrt{\frac{1}{N} \sum_{j=1}^{N} \left[ Z'(S_j) - \left( \sum_{j=1}^{N} Z'(S_j)/N \right) \right]^2} \tag{3}$$

where $Z'(S_j)$ are estimated values, $Z(S_j)$ observations at validation points and $N$ is the number of validation points.

$$\text{RMSE} = \sqrt{\frac{1}{N} \cdot \sum_{j=1}^{N} \left[ Z(S_j) - Z'(S_j) \right]^2} \tag{4}$$

$$\text{RMSE}_r = \frac{\text{RMSE}}{S_z} \tag{5}$$

where $S_z$ is the total variation (standard deviation).

The relationship between bacterial groups and physicochemical characteristics were explored with a canonical analysis of principal coordinates (CAP) in the Vegan package

(*Oksanen et al., 2017*). All measured soil characteristics, that is, pH, water content, EC, particle size distribution (sand, clay and loam), and organic and inorganic C were included in the CAP analysis. A principal component analysis was done using the packages FactoMineR (*Le, Josse & Husson, 2008*) while heatmaps were made using ComplexHeatmap (*Gu, Eils & Schlesner, 2016*). The Spearman test was used to calculated correlations between taxonomic groups and the soil characteristics (*Harrell, 2016*; *Wei & Simko, 2016*). The corSelect and multicol analysis in the Fuzzysim package (*Barbosa, 2016*) and BIOENV in Vegan package in R were also used to identify the variables with the least multicollinearity, those that showed direct relationships with communities and finally those that combined affected most the taxonomic groups.

## Data accessibility

Sequences obtained in this study were submitted to the NCBI Sequences Read Archive associated with the BioProject ID PRJNA414475 under the accession numbers: (SAMN07840001–SAMN07840013) for Archaea and (SAMN07840024–SAMN07840035) for Bacteria.

# RESULTS

## Soil characteristics

The pH varied between 10.3 and 10.6, while the soil WC between 13.3% and 55.5% (Table S1). The EC was highly variable along the transect and ranged from 7.7 to 179.8 dS m$^{-1}$ as was the organic C, which ranged from 4.5 to 26.3 g kg$^{-1}$ dry soil. The particle size distribution showed less variation and the soil texture ranged from sandy clay loam to loamy clay. Normality tests showed that each parameter was normally distributed without any tendency (https://doi.org/10.6084/m9.figshare.6357344.v1 for descriptive statistics table and https://doi.org/10.6084/m9.figshare.6349547 for Q–Q plots).

As was mentioned in the site description, some parts of the former Texcoco lake bed have been drained, which contributed to variations in salt content (*Dendooven et al., 2010*). Concordantly, the WC was highly variable upon soil sampling. The particle size distribution was more constant, although large variations in the clay content have been found in earlier studies (*Dendooven et al., 2015*). The organic C content was also highly variable and ranged from an organic poor soil (4.5 g C kg$^{-1}$) to an organic rich one (26.3 g kg$^{-1}$). The combination of variable plant material deposition and mineralization generated the large variability in soil organic matter content.

## Archaeal community structure

The largest Shannon diversity index was found in sample Tx-007 (5.87) and the lowest in sample Tx-008 (2.10), while the highest Chao1 richness in sample Tx-011 (300) and the lowest in sample Tx-012 (30) (Table 1). The PD ranged from 1.58 in sample Tx-012 to 3.82 in sample Tx-007. Diversity indexes were positively significantly correlated with the soil WC (Shannon $r = 0.74$) and the inorganic C content (Shannon $r = 0.71$), and significantly negatively correlated (Shannon $r = -0.57$) with the silt content ($p < 0.05$).

**Table 1 Archaeal diversity.** Diversity indices of the different taxonomic levels of Archaea at each sampling location (Tx001-Tx013) along a southeast transect in soil of the former lake Texcoco.

| Sample | Tx001 | Tx002 | Tx003 | Tx004 | Tx005 | Tx006 | Tx007 | Tx008 | Tx009 | Tx010 | Tx011 | Tx012 | Tx013 | Total |
|---|---|---|---|---|---|---|---|---|---|---|---|---|---|---|
| Sequences[a] | 802 | 1,439 | 571 | 160 | 837 | 723 | 1,365 | 1,225 | 802 | 1,112 | 826 | 630 | 927 | 11,419 |
| OTU$_{97}$[c] | 743 | 1,292 | 518 | 152 | 775 | 685 | 1,144 | 1,187 | 750 | 1,011 | 785 | 623 | 868 | 10,533 |
| Phylum | 2 | 1 | 1 | 2 | 1 | 2 | 2 | 2 | 2 | 1 | 2 | 2 | 1 | 2[b] |
| Class | 3 | 2 | 2 | 4 | 2 | 3 | 3 | 2 | 2 | 2 | 2 | 3 | 3 | 4 |
| Order | 3 | 2 | 2 | 4 | 2 | 4 | 3 | 2 | 2 | 2 | 2 | 3 | 3 | 5 |
| Family | 3 | 1 | 1 | 2 | 2 | 4 | 2 | 2 | 2 | 1 | 2 | 2 | 1 | 5 |
| Genera | 6 | 5 | 5 | 3 | 3 | 5 | 6 | 3 | 6 | 4 | 5 | 2 | 4 | 9 |
| Diversity parameters | | | | | | | | | | | | | | |
| Shannon | 5.24 | 5.15 | 5.01 | 5.61 | 5.33 | 4.79 | 5.87 | 2.10 | 5.66 | 4.72 | 5.31 | 2.87 | 5.60 | |
| Simpson | 0.95 | 0.94 | 0.94 | 0.97 | 0.94 | 0.91 | 0.97 | 0.49 | 0.97 | 0.91 | 0.94 | 0.70 | 0.96 | |
| Simpson E | 0.33 | 0.27 | 0.28 | 0.48 | 0.25 | 0.20 | 0.48 | 0.07 | 0.45 | 0.20 | 0.25 | 0.14 | 0.39 | |
| Simpson reciprocal | 21.26 | 17.42 | 16.70 | 33.03 | 17.11 | 11.47 | 39.03 | 1.95 | 32.15 | 11.37 | 17.55 | 3.29 | 28.54 | |
| Chao1 | 211.9 | 159.1 | 113.1 | 147.8 | 132.1 | 105.5 | 259.6 | 56.6 | 162.1 | 180.0 | 299.7 | 30.0 | 162.4 | |
| Good's coverage | 0.67 | 0.67 | 0.73 | 0.68 | 0.68 | 0.74 | 0.55 | 0.87 | 0.66 | 0.70 | 0.62 | 0.94 | 0.63 | |
| Phylogenetic diversity | 2.89 | 2.83 | 2.61 | 3.39 | 3.24 | 3.21 | 3.82 | 1.79 | 3.27 | 2.20 | 3.63 | 1.58 | 3.15 | |
| ACE[d] | 239.55 | 218.11 | 165.55 | 175.18 | 164.38 | 145.56 | 355.46 | 86.20 | 182.63 | 216.98 | 257.60 | 31.56 | 228.47 | |

**Notes:**
[a] after quality filtration.
[b] assigned to that taxonomic level.
[c] number of OTUs found before the subset was normalized to the sample with the lowest sequence count.
[d] ACE, Abundance-based Coverage Estimator.

Overall, sequences belonged to two archaeal phyla, four classes, five orders, five families and nine genera (Table 1). The number of archaeal phyla in each sample ranged from one to two, classes from two to four, orders from two to four, families from one to four and genera from three to six. Phylotypes belonging to the Euryarchaeota (mostly Halobacteriales) were the most abundant, but in two samples (Tx-008 and Tx-012) the relative abundance of Thaumarchaeota (mostly Nitrososphaerales) was greater (Fig. S1). Members of the Halobacteriaceae were found in each soil sample and were often the most abundant, with a relative abundance that ranged from 15.8% in sample Tx-012 to 98.7% in sample Tx-002. The Nitrososphaeraceae (mostly *Candidatus* Nitrososphaera) dominated in sample Tx-008 with a relative abundance of 78.4%. Phylotypes could be assigned to eight archaeal genera with *Candidatus* Nitrososphaera, *Cenarchaeum* and *Natronomonas* the most abundant.

The relative abundance of *Cenarchaeum* was positively significantly correlated with sand content ($r = 0.76$) and that of *Natronomonas* negatively ($r = -0.67$) with EC ($p < 0.01$) (Fig. 1A). The CAP grouped most soil samples together in the upper and lower left quadrant, except for samples Tx-008 and Tx-012 (Fig. 1B). Samples Tx-008 and Tx-012 were characterized by a larger value for dimension1 (CAP1), that is, a high relative abundance of *Candidatus* Nitrososphaera, a high silt content and a lower pH, EC and WC. The other soil samples were separated by differences in the relative abundance of most other archaeal genera, for example, *Haloterrigena* and *Halostagnicola*,

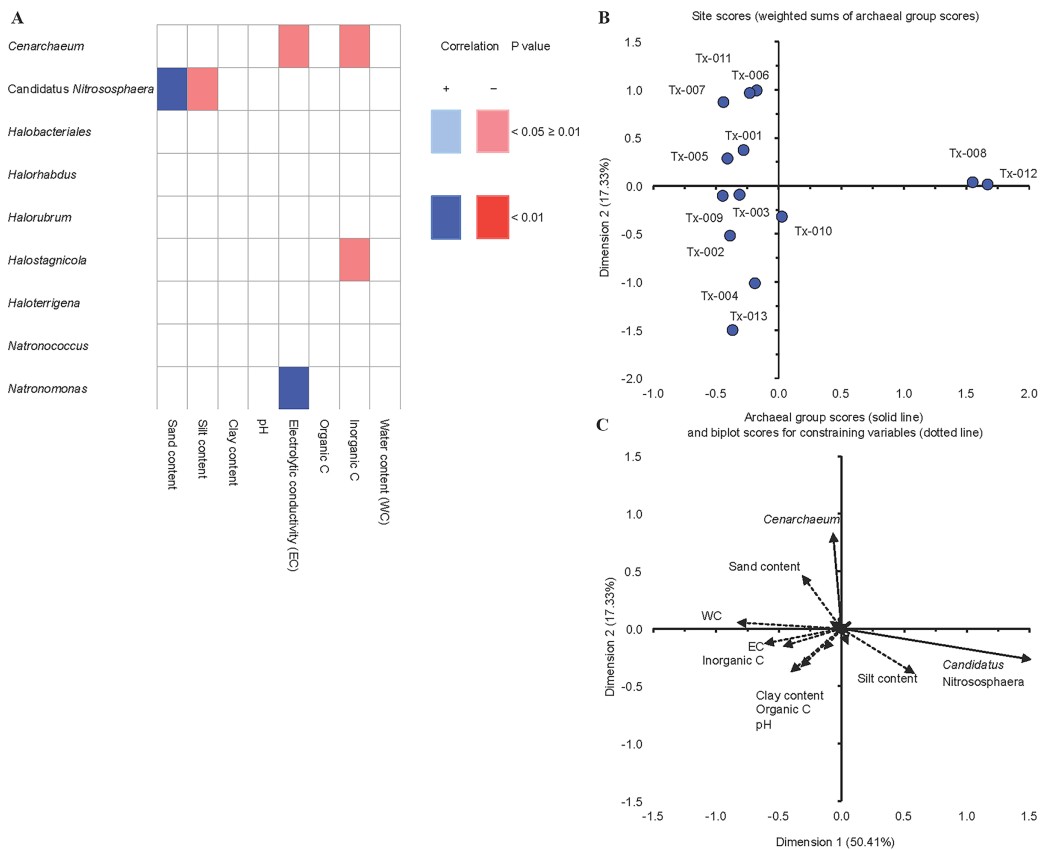

**Figure 1 Correlations between soil characteristics and archaeal groups.** (A) Heatmap with Spearman correlations between archaeal genera and soil characteristics. Positive significant correlations are marked in light blue ($p < 0.05$) and highly significant in dark blue ($p < 0.01$) while negative significant correlations are marked in light red ($p < 0.05$) and highly negative significant in dark red ($p < 0.01$). (B) Canonical analysis of principal coordinates (CAP) site scores (weighted sums of Archaeal group scores), represented as a solid line and (C) Biplot scores for group scores and constraining variables, represented as dotted line. Dimension 1 explained 50.41% of variability and Dimension 2 17.33%.

and differences in soil characteristics, such as organic and inorganic C, clay and sand content.

According to BIOENV analysis, a combination of WC, pH and clay content was the best subset to explain the archaeal community composition (Table S4).

## Bacterial communities structure

Alpha diversity parameters indicated that sample Tx-007 had the highest diversity (highest Shannon index) (Table 2). The Chao1 richness estimator ranged from 70 to 349.14, while the PD index from 4.95 to 7.37. Sample Tx-009 showed the highest PD. No significant correlation was found between the alpha diversity parameters and the measured soil characteristics. The estimated number of bacterial species was higher in sample Tx002 than in sample Tx012 based on both Chao1 and the ACE species richness estimators. *Kalwasinska et al. (2017)* also reported large differences in the number of estimated species in different layers of a saline soda lime sediment in Poland. The Shannon and Simpson

**Table 2 Bacterial diversity.** Diversity indices of the different taxonomic levels of Bacteria at each sampling location (Tx001-Tx013) along a southeast transect in soil of the former lake Texcoco.

| Sample | Tx001 | Tx002 | Tx003 | Tx004 | Tx006 | Tx007 | Tx008 | Tx009 | Tx010 | Tx011 | Tx012 | Tx013 | Total |
|---|---|---|---|---|---|---|---|---|---|---|---|---|---|
| Sequences[a] | 668 | 676 | 436 | 323 | 231 | 794 | 304 | 700 | 297 | 245 | 280 | 167 | 5,121 |
| $OTU_{97}$[c] | 511 | 558 | 346 | 309 | 217 | 706 | 266 | 641 | 278 | 220 | 232 | 155 | 4,439 |
| Phylum | 11 | 9 | 6 | 11 | 8 | 12 | 11 | 14 | 9 | 9 | 8 | 10 | 17[b] |
| Class | 16 | 18 | 11 | 20 | 14 | 21 | 19 | 22 | 16 | 17 | 21 | 21 | 43 |
| Order | 14 | 15 | 12 | 18 | 15 | 17 | 17 | 19 | 15 | 18 | 18 | 21 | 44 |
| Family | 14 | 15 | 12 | 15 | 14 | 18 | 19 | 19 | 14 | 15 | 16 | 17 | 55 |
| Genera | 9 | 13 | 8 | 12 | 12 | 13 | 10 | 10 | 8 | 6 | 17 | 9 | 41 |
| Diversity indexes | | | | | | | | | | | | | |
| Shannon | 5.52 | 5.92 | 5.96 | 6.12 | 5.74 | 6.21 | 5.72 | 6.11 | 5.52 | 5.75 | 5.57 | 6.13 | |
| Simpson | 0.96 | 0.98 | 0.97 | 0.98 | 0.98 | 0.98 | 0.97 | 0.98 | 0.97 | 0.98 | 0.97 | 0.98 | |
| Simpson E | 0.40 | 0.54 | 0.48 | 0.66 | 0.64 | 0.71 | 0.58 | 0.71 | 0.47 | 0.61 | 0.60 | 0.78 | |
| Simpson reciprocal | 27.08 | 42.34 | 38.77 | 54.82 | 42.81 | 61.27 | 39.16 | 57.23 | 29.99 | 41.67 | 35.11 | 61.27 | |
| Chao1 | 323.00 | 349.14 | 207.43 | 178.06 | 130.08 | 197.24 | 93.25 | 168.35 | 132.33 | 129.50 | 70.00 | 126.00 | |
| Good's coverage | 0.59 | 0.50 | 0.52 | 0.53 | 0.67 | 0.50 | 0.71 | 0.56 | 0.67 | 0.66 | 0.82 | 0.62 | |
| Phylogenetic diversity | 6.36 | 6.15 | 5.88 | 6.24 | 4.97 | 7.11 | 6.10 | 7.37 | 5.16 | 4.95 | 5.40 | 6.86 | |
| ACE[d] | 222 | 337 | 223 | 227 | 133 | 224 | 128 | 182 | 138 | 144 | 74 | 131 | |

Notes:
[a] after quality filtration.
[b] assigned to that taxonomic level.
[c] number of OTUs found before the subset was normalized to the sample with the lowest sequence count.
[d] ACE, abundance-based coverage estimator.

reciprocal indices were the lowest in Tx001 and Tx010 samples. Sample Tx010 had the highest EC value, so the extreme hypersalinity affected the bacterial diversity.

Overall phylotypes belonged to 20 bacterial phyla, 48 classes, 57 orders, 68 families and 49 genera. The most abundant phyla in the samples were Actinobacteria followed by Proteobacteria and Gemmatimonadetes (Fig. S2). Chlorobi and Firmicutes showed the largest variability in relative abundance along the transect. Members of *Euzebya* (Actinobacteria) were the most abundant, followed by KSA1 (Bacteroidetes) and *Bacillus* (Firmicutes) (Fig. S3).

The relative abundance of Chlorobi was significantly negatively correlated with the loam content ($r = -0.74$) and positively with WC ($r = 0.73$) ($p < 0.01$), while Spirochaetes were significantly and positively correlated with inorganic C ($r = 0.62$, $p < 0.05$) (Fig. 2A). The WC was significantly correlated with the relative abundance of Chlorobi ($r = 0.73$, $p < 0.05$) and Proteobacteria ($r = -0.68$, $p < 0.05$). Chlorobi were correlated significantly with most of soil characteristics, positively with sand and inorganic C content and negatively with loam content (Fig. 2A) ($p < 0.05$). The relative abundance of the genus B-42 was significantly and negatively correlated with organic C ($r = -0.68$), KSA1 positively ($r = 0.89$) with WC and *Methylonatrum* positively ($r = 0.54$) with inorganic C ($p < 0.01$) (Fig. 2B). The CAP of the bacterial phyla and the soil characteristics separated soil samples Tx-008 and Tx-012 from the other samples (Fig. 3). Samples Tx-008 and Tx-012 were characterized by a larger relative abundance of Proteobacteria (Fig. 3A).

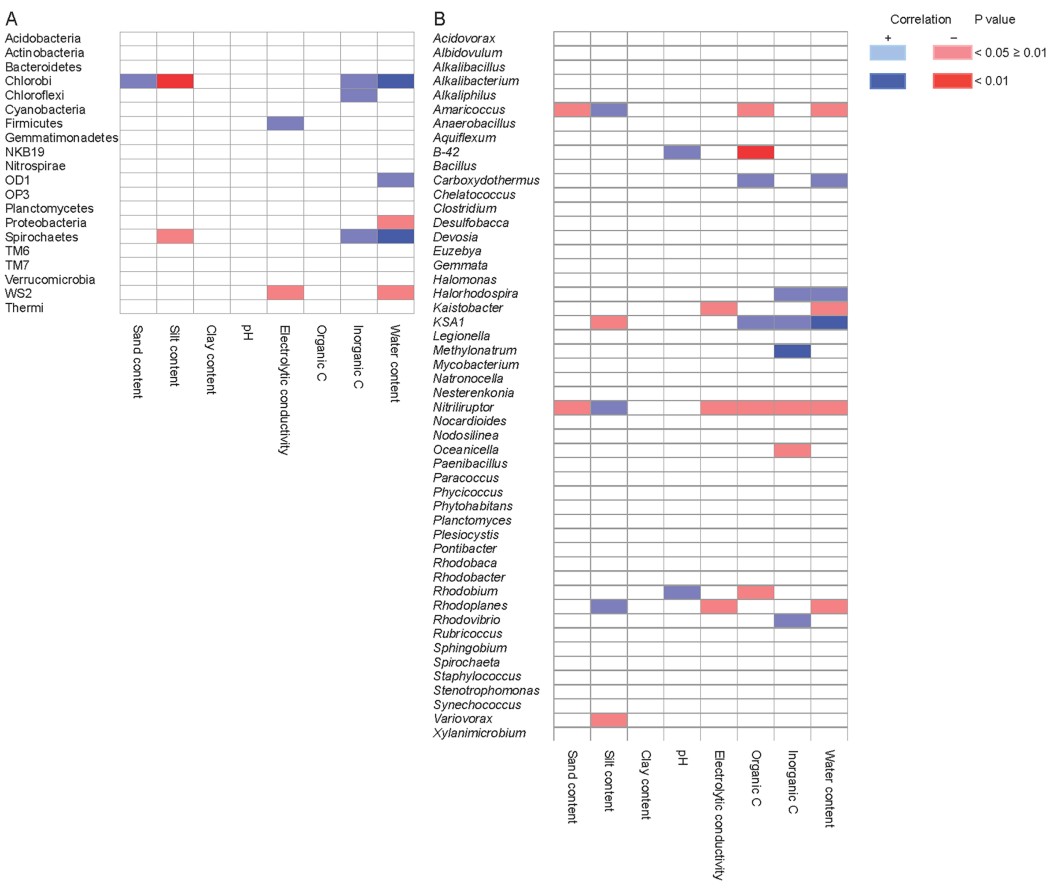

**Figure 2 Correlation between bacterial groups and soil characteristics.** Heatmap with Spearman correlations between (A) bacterial phyla or (B) genera and selected soil characteristics. Positive significant correlations are marked in light blue ($p < 0.05$) and highly significant in dark blue ($p < 0.01$) while negative significant correlations are marked in light red ($p < 0.05$) and highly negative significant in dark red ($p < 0.01$).

The CAP analysis with the bacterial genera showed that samples Tx-008 and Tx-012 had a larger relative abundance for *Amaricoccus*, *Kaistobacter*, *Paracoccus* and *Pontibacter* than the other samples (Fig. 3D). Generally, the combination of WC, EC and silt content were the parameters that best correlated with the relative abundance of the bacterial phyla (Table S4).

The ratios between the different alpha diversity indexes of Archaea/Bacteria were calculated. In Tx008, the diversity of Bacteria was higher than that of Archaea (Shannon index ratio 0.37) and the bacterial richness was larger than that of Archaea in Tx012 (0.43 Chao1 and ACE). In Tx011, the archaeal richness was larger than that of Bacteria (2.31 for the Chao1 index and 1.70 for the ACE index). The locations Tx001, Tx004, Tx007 and Tx009 had a similar diversity for Archaea and Bacteria.

This dominance of Bacteria over Archaea was detected in sites with the lowest EC, while the opposite was observed in sites with high EC. The Spearman correlation for the ratio of archaeal versus bacterial diversity was positive with WC ($r = 0.86$, $p < 0.05$) and inorganic C ($r = 0.67$, $p < 0.05$), and negative with the loam content ($r = -0.81$, $p < 0.05$).
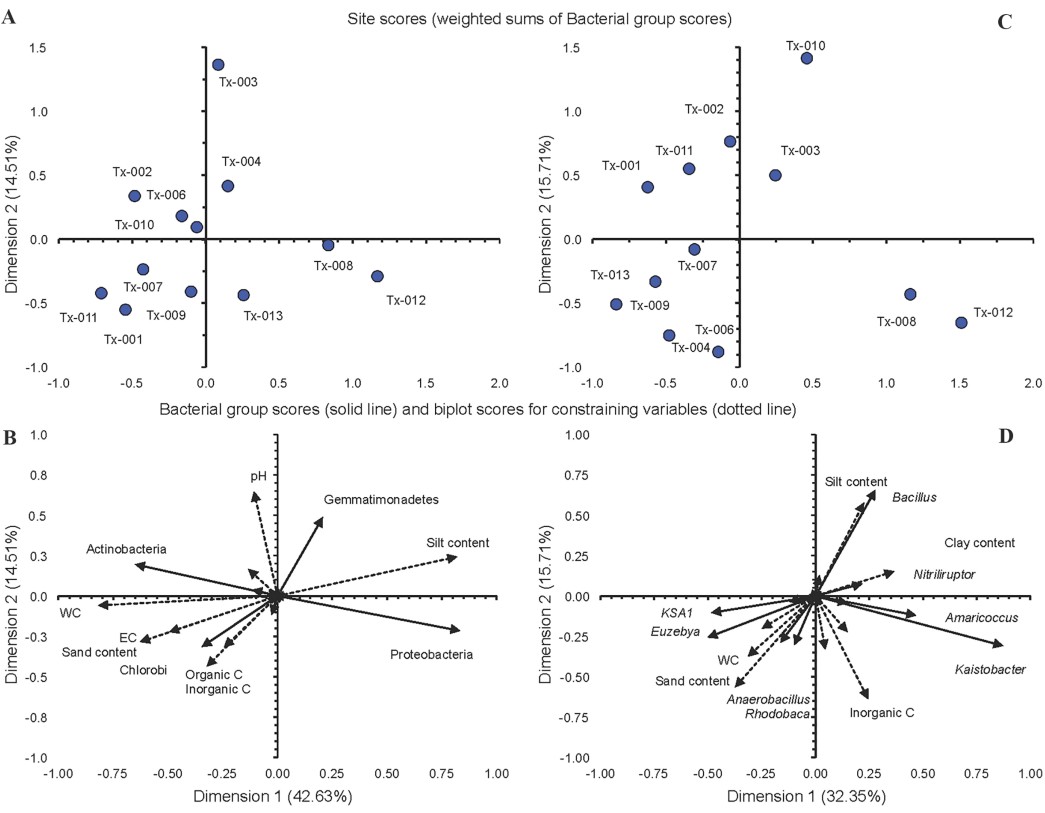

**Figure 3 Canonical analysis of principal coordinates of bacterial groups.** Canonical analysis of principal coordinates (CAP) with site scores (weighted sums of Bacterial group scores) and biplot scores for (A) and (B) bacterial phyla (Dimension 1 explained 42.63% of the variation and dimension 2 14.51%), (C) and (D) bacterial genera (dimension 1 explained 32.35% and dimension 2 15.71% of the variation. Bacterial groups scores are represented as a solid line and biplot scores for constraining variables as a dotted line.

A positive correlation was found for the ratio of archaeal versus bacterial richness with EC ($r = 0.73$, $p < 0.05$).

## Geographical situation of physicochemical parameters

The transect followed a diagonal orientated from Northwest to Southeast with some salt crusts and patches of vegetation. Mantel autocorrelograms showed significantly positive and negative spatial autocorrelations at certain distance classes along the transect thus describing a spatial periodic structure ($p < 0.05$). Cloud semivariograms for the physicochemical factors did not show atypical values or tendencies. For the pooled variograms, the model that best fitted the empirical semivariogram of each soil characteristic was periodic, that is, with presence of patches. The calculated sill, rank, nugget and ratio nugget/sill values of each parameter are given in Table S2. Organic and inorganic C content, clay, silt, WC and EC showed a high spatial autocorrelation (nugget/sill ratio < 0.25) while pH and sand showed a medium spatial dependence (nugget/sill ratio between 0.25 and 0.75). Soil characteristics have a high spatial autocorrelation if the nugget/sill ratio < 0.25, medium if between 0.25 and 0.75, and low > 0.75 (*Cambardella et al., 1994*).

## Interpolated mapping of physicochemical factors

The studied soils were characterized as alkaline and defined as mainly sandy clay loam. Low pH values (<10.2) were predicted in a delimited area of approximately 180 m$^2$, located in the south-eastern part of the polygon, but the variation of pH values across the predicted area were low. The predicted EC values were the lowest (<7 dS m$^{-1}$) in the south-eastern part of the polygon and for a 300 m$^2$ area near to the center of the sampling transect (501110–501155 Easting and 2157425–2157475 Northing). These areas with low EC were visually correlated with the vegetation patch and with a high abundance of Thaumarchaeota. The predicted WC was more heterogeneous distributed spatially with values that ranged from 10% to 70%. The lowest WC was predicted in the same south-eastern part of the polygon and coincided with the lowest EC (around site Tx-012). The highest WC values (70%) were predicted at the north and south edges of the polygon. Inorganic C content was homogeneous across the sampling transect as the standard deviation of the interpolated parameters was low (1.95). The organic C content showed more variation (ranged from 5 to 25 g Kg$^{-1}$) with the highest values in the southeast of the polygon and the lowest around Tx008. Cross validation showed that some places (8–16%) were over- or underestimated for almost all physicochemical parameters. The ASE and RMSE were similar for pH and WC, but not for EC, and inorganic C, clay and silt content (Table S3). This confirmed variability in the predicted values for almost all measured soil characteristics.

The RMSE$r$ for WC was 1.108 and 1.378 for clay content, indicating that the model accounted for less than 50% of the variability at the validation points (Table S3). As such, the predictions are not fully satisfactory for these soil characteristics.

## Distribution models for Bacteria and Archaea

The ratio of the Shannon diversity of Archaea versus Bacteria across the predicted area ranged from zero to one. At the center of the surrounding transect, that is, around site Tx-007, a higher archaeal diversity was predicted (Fig. S4). Distribution of Thaumarchaeota showed two patches of high abundance within the prediction area that correlated negatively with the predicted abundance distribution of Euryarchaeota and negatively with predicted EC values (Fig. 4). Euryarchaeota were found along the whole prediction area with the lowest abundances in patches with high abundance of Thaumarchaeota. The richness and diversity prediction for Bacteria (Chao1 estimator and Shannon index) showed the highest values in patches around sites Tx-012 and Tx-006 where EC and pH were lowest. A higher relative abundance of Proteobacteria and a lower one for Actinobacteria was predicted around Tx-012 (Fig. 5) The ASE and RMSE were similar for most communities except for bacterial richness (Chao1). The RMSE$r$ was greater than 70% for archaeal phyla and some Bacteria indicating that the predictions are not fully satisfactory for those communities.

Sudden changes in the relative abundance of bacterial and archaeal groups were predicted, especially where EC and WC were lower. It is possible that transition zones of microbial distribution may be more easily observed at shorter distances than those considered in the present study.

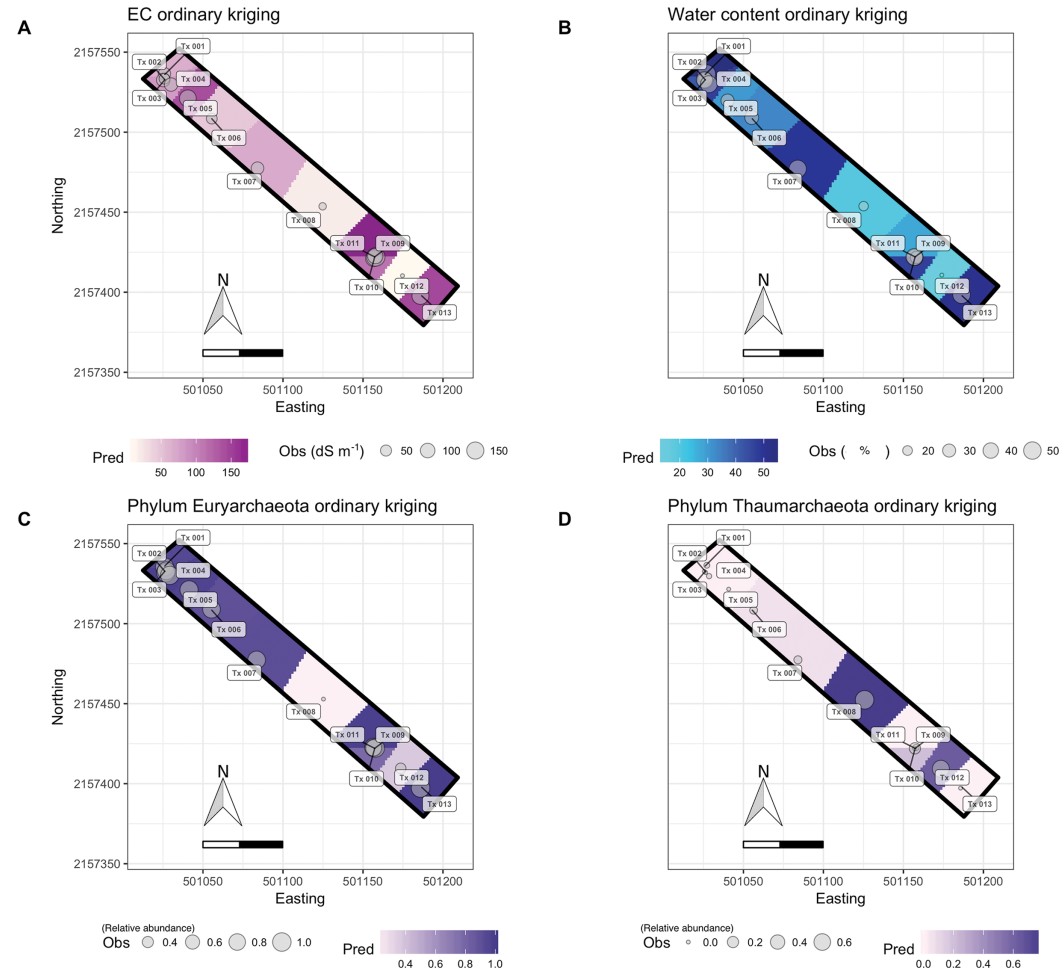

**Figure 4 Ordinary kriging maps of electrolytic conductivity, water content and archaeal phyla.** (A) Electrolytic conductivity (EC), (B) water content (WC), (C) relative abundance of Euryarchaeota and (D) relative abundance of Thaumarchaeota. Pred: predicted values by ordinary kriging on a 3,000-pixel grid (2.38 m$^2$ each). Obs: observed values at the southeast transect sampling sites (Tx001-Tx013). The coordinates are in UTM (14N). Full-size 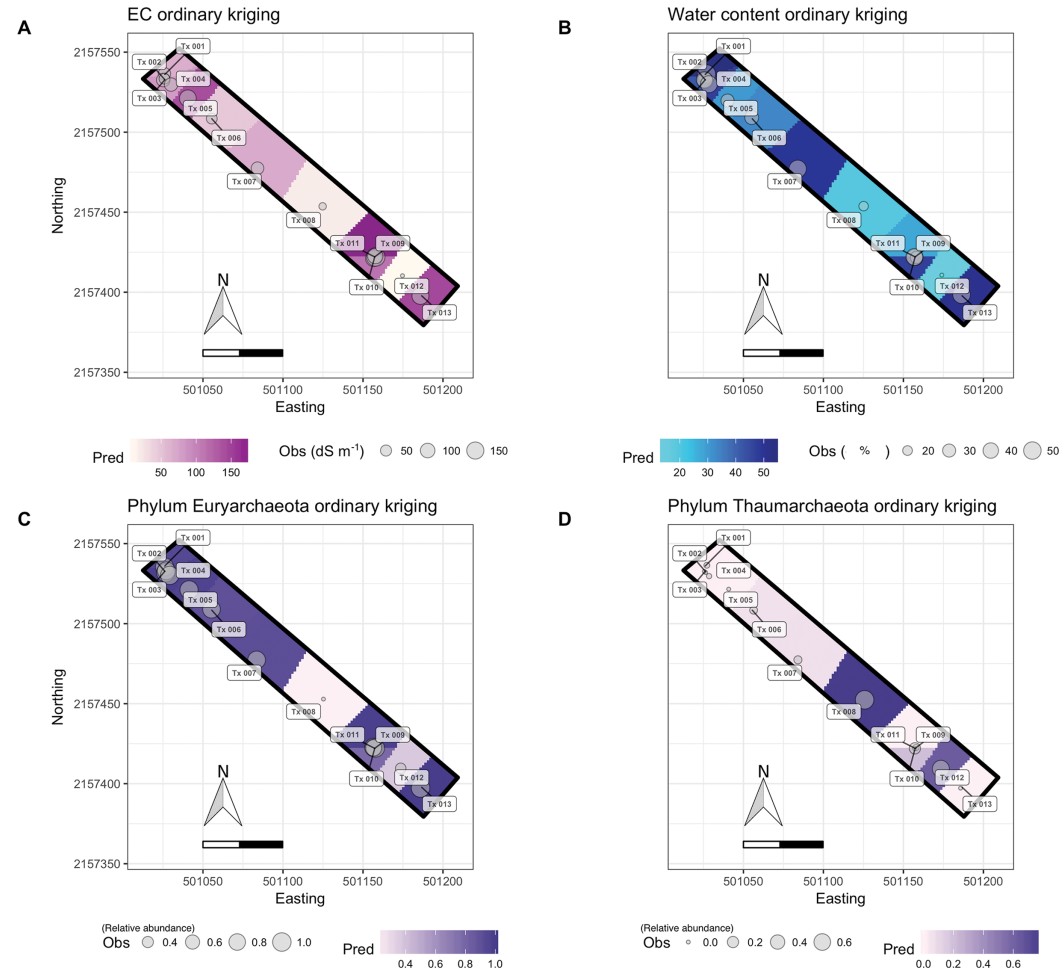 DOI: 10.7717/peerj.6127/fig-4

## DISCUSSION

### Archaeal community structure

Euryarchaeota was the dominant phylum across the sampling transect except for soil with a lower EC (Tx008 and Tx012) where Thaumarchaeota dominated. These results are consistent with other studies of alkaline and hyper-saline soils and sediments, where Euryarchaeota are the dominant phylum, with Halobacteriales the most abundant order (*Hollister et al., 2010*; *Ma & Gong, 2013*; *Budakoglu et al., 2014*; *Vogt et al., 2017*). Euryarchaeota were often the most abundant in the Texcoco soil (*Valenzuela-Encinas et al., 2012*; *Navarro-Noya et al., 2015*) although Thaumarchaeota dominated in soils with lower EC, pH and clay content (*Wessén et al., 2011*; *Hatzenpichler, 2012*; *Webster et al., 2015*).

The lowest values of archaeal diversity and richness along the transect were found in the sites Tx008 and Tx012. This could be explained by the abundance of Thaumarchaeota

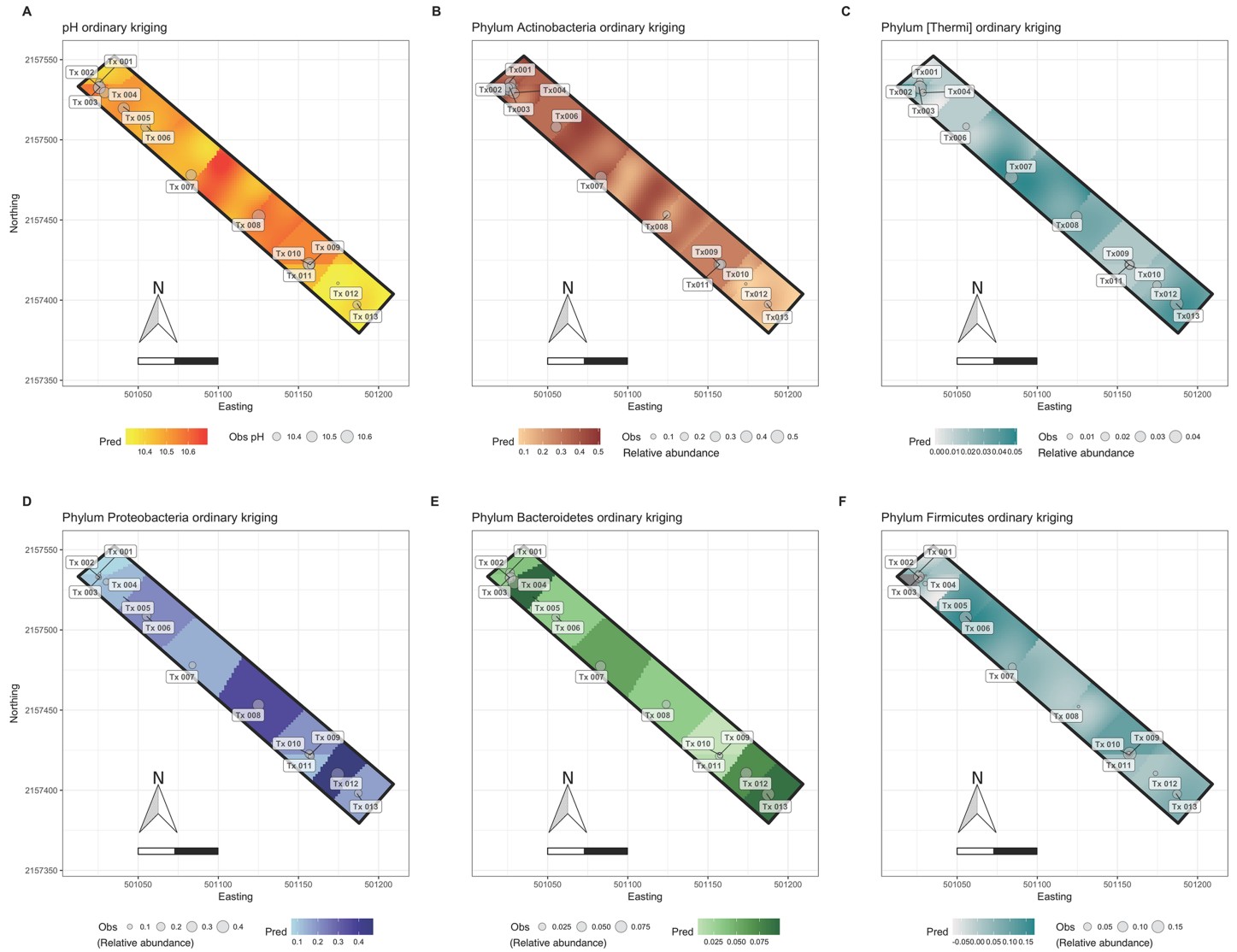

**Figure 5 Ordinary kriging maps for pH and some bacterial phyla.** (A) pH, (B) Actinobacteria, (C) (Thermi), (D) Proteobacteria, (E) Bacteroidetes, (F) Firmicutes. Pred: predicted values by ordinary kriging on a 3,000-pixel grid (2.38 m$^2$ each); Obs: observed values at the southeast transect sampling sites (Tx001-Tx013). The coordinates are in UTM (14N).

and the lowest values of salinity at these sites and, probably the most oligotrophic conditions along the transect. Similar results have been obtained in other studies in which it was found that soil dominated by Thaumarchaeota had a lower diversity than the samples where other archaeal phyla dominated (*Auguet & Casamayor, 2013*; *Navarro-Noya et al., 2015*). The greatest diversity was found in the sites Tx009 and Tx007, and the latter showed the greatest uniformity of species (Simpson E Index). On the other hand, the greatest richness was found in site Tx011.

Numerous studies have found a strong effect of soil properties on soil microbial communities (*Bru et al., 2011*; *Shange et al., 2013*; *Pasternak et al., 2013*; *Sullivan, Mcbride & Thies, 2013*). For instance, archaeal ammonium oxidizers are affected by oxygen and ammonia concentrations, temperature, pH and salinity (*Duff, Zhang & Smith, 2017*).

*Bru et al. (2011)* found an effect of pH on the crenarchaeal ammonia oxidizers, while *Shange et al. (2013)* a strong correlation between bacterial phyla, such as Bacteroidetes, Proteobacteria and Actinobacteria, and substrate availability along a land use gradient. *Pasternak et al. (2013)* reported that abundances of microbial communities were significantly correlated with soil characteristics, such as texture, organic matter and water content. *Sullivan, Mcbride & Thies (2013)* found a large effect of pH, total soil S, extractable soil Cu and Zn on the soil microbial community structure. In this study, *Cenarchaeum,* a chemolithoautotrophic mesophilic aerobic oxidizer of $NH_4^+$ (*Hatzenpichler, 2012*; *Banerjee & Siciliano, 2012*) was favoured by clayey soil whereas *Natronomonas* was favoured by high EC values. In this study *Candidatus* Nitrososphaera, also an oxidizer of ammonium, dominated in soil with lower EC as well as higher silt content. Both *Cenarchaeum* and *Candidatus* Nitrososphaera (Thaumarchaeota), Archaea with a potential to oxidize ammonium, have been found in hypersaline soils and marine environments (*Hatzenpichler, 2012*; *Tolar, King & Hollibaugh, 2013*; *Navarro-Noya et al., 2015*). Members of Thaumarchaeota have been detected in oligotrophic sites and have been correlated with a lower pH and salt content, but especially with low concentrations of ammonium (*Wessén et al., 2011*; *Marusenko et al., 2013*; *Bollman, Bullerjahn & Mckay, 2014*).

Overall, archaeal diversity was favored by soil moisture, total and inorganic C content, and was negatively correlated with the silt content. *Hollister et al. (2010)* separated the archaeal alpha diversity of hypersaline sites based mainly on the WC, that is, distinguishing between terrestrial and aquatic sites. Although they found a positive correlation of archaeal abundance with EC and salinity, the most influential variable to explain differences was WC. The archaeal abundance increased directly with WC and Archaea were almost absent from the terrestrial sites of the sampling transect. Similar results were found in this study, as the archaeal spatial structure was driven mostly by WC.

## Bacterial community structure

The bacterial community composition was less controlled by soil characteristics than the archaeal composition in this study, as the variation in the relative abundance of most phyla was low. Proteobacteria, Bacteroidetes and Actinobacteria are the dominant soil bacterial taxa in saline and hypersaline soils (*Ma & Gong, 2013*; *Canfora et al., 2014*; *Liu et al., 2014*; *Santini, Warren & Kendra, 2015*) as found in this study. Gemmatimonadetes detected previously in hyper-saline soils (*Canfora et al., 2014*) and appeared not to be affected by EC. This might be because some members of Gemmatimonadetes can adapt to a wide range of different conditions and some species are not affected by salinity (*Ma & Gong, 2013*). Chlorobi, often one of the minor phyla, has been related frequently to organic C in other saline and soda sites, but also in arable soils (*Constancias et al., 2015*). In this study, the relative abundance of Chlorobi showed the largest variability along the transect and was negatively correlated with the loam content and positively with WC. The relative abundance of Firmicutes was also highly variable, but not controlled by soil characteristics in this ecosystem. Firmicutes are ubiquitous in soil and can be found in most ecosystems (*Liu et al., 2014*; *Santini, Warren & Kendra, 2015*).

Different factors, such as WC, salt content, pH and organic material, control the bacterial community structure. *Wasserstrom et al. (2017)* found that WC affected the bacterial community composition in a coastal dune. Microbial structure is affected by fluctuations in WC in both extreme dry or wet conditions. Microorganisms need water in their environment for their normal metabolic functioning and although some species have been found in extreme dry conditions, their activity normally stops when the WC drops below a critical amount (*Crits-Christoph et al., 2013*). In wet conditions, oxygen fluxes are impeded and microbial activity changes from aerobic to anaerobic, which ultimately changes bacterial community composition. Some microorganisms are strictly aerobic, so their activity is inhibited while facultative and strict anaerobes flourish. Consequently, WC will control microbial community and more so in a salt rich soil where osmotic pressures fluctuate more with changing WC than in a salt poor soil. For instance, *Hollister et al. (2010)* reported that the abundance of Archaea and Bacteria was correlated to a greater extent with WC than with EC or $Na^+$ content.

It has been reported that pH is an important driver of microbial communities distribution at large scales when a gradient is present (*Rousk et al., 2010*; *Liu et al., 2014*; *Constancias et al., 2015*). It has not been determined yet, whether the effect of pH is direct as a physiological barrier, or indirectly through other related chemical factors. On a local scale, as in this study, pH did not seem to be the main distribution driver as its variation through the sampling transect was small, that is, 10.3–10.6. This is consistent with the findings of microbial communities distribution in hypersaline soils (*Hollister et al., 2010*; *Canfora et al., 2014*) and the extreme dry Atacama desert (*Crits-Christoph et al., 2013*).

*Methylonatrum* was first described in hypersaline lakes in Russia by *Sorokin et al. (2007)*. They described it belonging to the Gammaproteobacteria that was able to grow on $C_1$ substrates. Classified as a methylotroph genus, *Methylonatrum* can grow under salt saturation and alkaliphilic conditions, whose optimal pH reaches 10.5. In the present study, this genus showed a high correlation with the C inorganic content, and was detected at pH 10.6, which is consistent with the findings of *Simachew et al. (2015)*. They found *Methylonatrum* only in an intermediate saline site (25%) dominated by Archaea.

KSA1 and B-42, the second and third most abundant genus in this study, are little known bacterial groups. They were found for the first time in lake sediments comparable to soil of the former lake Texcoco. The candidate division KSA1, the second most abundant genus found in this study, was reported for the first time in a hypersaline sulphide-rich black mud environment in a marsh (*Tanner et al., 2000*) and was classified as belonging to Bacteroidetes (*McDonald et al., 2012*). Members of B-42 were found in freshwater sediments and activated sludge from a seawater-processing wastewater treatment plant (*Sánchez et al., 2011*; Genbank accession number: FN598000; *Hamamura et al., 2014*; accession number: KC852965.1; *Chen et al., 2015*). They were also found in creosote-polluted soils (*Tejeda-agredano et al., 2013*; accession number: JQ771979.1).

Overall the alpha diversity for bacterial communities was higher than that of Archaea. It differs from other studies in hypersaline and saline environments where the diversity of Archaea was larger than that of Bacteria (*Canfora et al., 2015*) and was more

accentuated for archaeal ammonia oxidizers (*Wessén et al., 2011*; *Marusenko et al., 2013*; *Bollman, Bullerjahn & Mckay, 2014*). The richness of Archaea was greater in some sites (Tx-007, Tx-010, Tx-011 and Tx-013) with extreme soil physicochemical characteristics (i.e., high EC and pH values) as Archaea are generally better adapted to extreme halophilic environments (*Oren, 2014*).

A low number of sequences have been reported in similar saline environments. For instance, *Kalwasinska et al. (2017)* reported only 70 archaeal sequences in an extreme saline soil and only 321 bacterial OTUs, similar to those found in this study. *Ma & Gong (2013)* reported only 1,039 sequences for Archaea (602 OTUs assignations) after a meta-analysis of all reported sequences in different saline and hyper-saline soil environments. *Vogt et al. (2017)* using Illumina as a sequencing platform, obtained a large number of sequences in hypersaline mats (up to 25,239 for Bacteria and 47,606 for Archaea). However, they were only able to assign 221-440 OTUs for Bacteria and 138-193 archaeal OTUs. The richness indices they reported were similar to those found in this study.

The high proportion OTU/sequences might be due to a combination of factors, that is, the complexity of the sample, the ambiguity of hypervariable sites of the 16S rRNA gen, and the alignment and clustering method used (*Konstantinidis & Tiedje, 2005*; *Chen et al., 2013*; *Li et al., 2014*; *Rideout et al., 2014*; *He et al., 2015*; *Nguyen et al., 2016*).

## Spatial patterns and microbial distribution

An attempt was made to determine changes in microbial communities as defined by soil characteristics, describing spatial distribution patterns along a sampling transect. Previous studies of the extreme heterogeneous former lakebed at Texcoco (*Valenzuela-Encinas et al., 2009*, *2012*; *Dendooven et al., 2010*; *Navarro-Noya et al., 2015*) provided indications of the scale and direction of sampling that was required to determine with geostatistical and multivariate analysis the relative influence of the measured soil characteristics on the microbial structure. From the previously conducted studies, the presence of gradients in soil pH and EC values along the sampling transect was known. On the selected scale, however, only a high variability of EC was found but a narrow pH range. Moreover, periodicities of water and organic C content were observed.

Many studies have found that pH is an important driver of bacterial composition, even when its variation is low (*Constancias et al., 2015*). However, in an extreme soil such as soil of the former lake Texcoco and on a local scale, the direct influence of pH on microorganisms is outweighed by other characteristics. The EC and WC showed a high variability along the sampling transect, and the pattern of irregular distribution observed for both showed a close relationship with abrupt changes in the Archaea communities. The vegetation patches in this extreme area are correlated with low EC values and a high abundance of Thaumarchaeota. Thaumarchaeota members have been detected in oligotrophic sites and found to correlate with a lower pH and salt content, especially at low ammonium concentrations (*Wessén et al., 2011*; *Marusenko et al., 2013*; *Bollman, Bullerjahn & Mckay, 2014*).

Despite the low sampling efforts, the bacterial distribution maps revealed a pattern with low variability along the sampling transect for the most abundant phylum, that is, Actinobacteria, Proteobacteria and Bacteroidetes (Figs. 5B, 5D and 5E). The highest abundances for Proteobacteria were observed in sites with lower EC values and the distribution of this phylum was correlated positively with WC. The low variability of these taxa might be due to that some members of these phyla are well adapted to salinity and alkalinity (*Ma & Gong, 2013*). The variability of less abundant phyla, such as Chlorobi and Firmicutes (Fig. 5F), was more variable with the abundance of Chlorobi correlated with most soil characteristics and Firmicutes with none.

The choice of scale is important to define spatial patterns. It has been observed that at a local scale, some factors show spatial homogeneity, as it was found with pH in this study. It differs from large scales studies, where pH can be more variable and consequently be the main driver for communities distribution, such as shown by *Navarro-Noya et al. (2015)*. Spatial variability of phyla at a local scale might be low as members of the same phylum can share similar characteristics and metabolic activities, whereas changes in structure and composition can be highlighted by an abundance of order/families (*Crits-Christoph et al., 2013*). Although temporal variations were not investigated in this study, a strong correlation in temporal and spatial scales has been reported (*Lauber et al., 2013*; *Regan et al., 2014*; *Wasserstrom et al., 2017*). Unfortunately, after the first sampling, it was not possible to sample the study site more intensively as it was assigned to be part of the new International Airport of Mexico City.

## CONCLUSION

The relative abundance of archaeal groups was controlled by soil characteristics. Thaumarchaeota were more abundant in locations with the lowest EC and WC, and the presence of *Distichlis spicata*. Consequently, abundances of Euryarchaeota were low in these points. Bacteria were less affected by physicochemical factors, but the presence or absence of bacterial groups was related to WC, pH and EC. At the scale of the present study, the main driving factor to explain the microbial community structure was WC. Ordinary kriging with some of the factors and response variables explained the variability in the sampling sites. For other factors and response variables, however, a more intensive sampling will be required. Kriging revealed abrupt changes for archaeal communities related with variability of WC and EC, but not for Bacteria. The relative abundance of most bacterial groups, for example, Actinobacteria and Proteobacteria, was less variable than that of most archaeal groups, but the relative abundance of others, for example, Chlorobi and Firmicutes, was highly variable. For instance, the relative abundance of the Firmicutes varied between <0.1% in Tx008 and 25.0% in Tx010. It was possible to map soil community distributions in extreme heterogeneous environment using a Geostatistical approach. The use of novel omic tools and Geostatistical analyses can determine the functionality in an area and help to design novel management practices for extreme ecosystems.

# ACKNOWLEDGEMENTS

The authors thank ABACUS (CONACyT) for providing time on the computer and "*Comisión nacional de agua* (CNA)" for access to the sampling site.

## Funding

This research was funded by 'Centro de Investigación y de Estudios Avanzados' (Cinvestav, Mexico), and 'Apoyo Especial para Fortalecimiento de Doctorado PNPC 2013, 2014' and project 'Infraestructura 205945' from 'Consejo Nacional de Ciencia y Tecnología' (CONACyT, Mexico). Martha Adriana Martínez-Olivas (230274) and Carmine Fusaro received a doctoral grant from CONACyT. The funders had no role in study design, data collection and analysis, decision to publish, or preparation of the manuscript.

## Grant Disclosures

The following grant information was disclosed by the authors:
Centro de Investigación y de Estudios Avanzados.
Apoyo Especial para Fortalecimiento de Doctorado: PNPC 2013, 2014 and project Infraestructura: 205945.
Consejo Nacional de Ciencia y Tecnología.
Martha Adriana Martínez-Olivas: 230274.
CONACyT.

## Competing Interests

The authors declare that they have no competing interests

## Author Contributions

- Martha Adriana Martínez-Olivas performed the experiments, analyzed the data, prepared figures and/or tables, authored or reviewed drafts of the paper, approved the final draft.
- Norma G. Jiménez-Bueno performed the experiments, authored or reviewed drafts of the paper, approved the final draft.
- Juan Alfredo Hernández-García analyzed the data, authored or reviewed drafts of the paper, approved the final draft.
- Carmine Fusaro analyzed the data, prepared figures and/or tables, authored or reviewed drafts of the paper, approved the final draft.
- Marco Luna-Guido performed the experiments, authored or reviewed drafts of the paper, approved the final draft.
- Yendi E. Navarro-Noya conceived and designed the experiments, contributed reagents/materials/analysis tools, prepared figures and/or tables, authored or reviewed drafts of the paper, approved the final draft.

- Luc Dendooven conceived and designed the experiments, analyzed the data, contributed reagents/materials/analysis tools, prepared figures and/or tables, authored or reviewed drafts of the paper, approved the final draft.

## Data Availability

Sequences obtained were submitted to the NCBI Sequences Read Archive associated with the BioProject ID PRJNA414475 under the accession numbers: (SAMN07840001–SAMN07840013) for Archaea and (SAMN07840024–SAMN07840035) for Bacteria.

## Supplemental Information

Supplemental information for this article can be found online at http://dx.doi.org/10.7717/peerj.6127#supplemental-information.

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

## FURTHER READING

**Bissett A, Richardson AE, Baker G, Wakelin S, Thrall PH. 2010.** Life history determines biogeographical patterns of soil bacterial communities over multiple spatial scales. *Molecular Ecology* **19(19)**:4315–4327 DOI 10.1111/j.1365-294X.2010.04804.x.

**Bivand RS, Pebesma EJ, Gomez-Rubio V. 2013.** *Applied spatial data analysis with R*. Second Edition. New York: Springer. *Available at* http://www.asdar-book.org/.

**Ettema CH, Wardle DA. 2002.** Spatial soil ecology. *Trends in Ecology & Evolution* **5347**:177–183.

**Gallardo A, Maestre FT. 2008.** Métodos geoestadísticos para el análisis de datos ecológicos espacialmente explícitos. Capítulo 6. In: Maestre FT, Escudero A, Bonet A, eds. *Introducción al análisis espacial de datos en ecología y ciencias ambientales: Métodos y aplicaciones*. Madrid: Dikynson, 215–272.

**Hanson CA, Fuhrman JA, Horner-Devine MC, Martiny JBH. 2012.** Beyond biogeographic patterns: processes shaping the microbial landscape. *Nature Reviews Microbiology* **10(7)**:497–506 DOI 10.1038/nrmicro2795.

**Hurst CJ. ed. 2016.** *Their world: A diversity of microbial environments.* Vol. 1. Ohio: Springer International Publishing Switzerland.

**Jeelani J, Kirmani NA, Sofi JA, Mir SA, Wani JA, Rasool R, Sadat S. 2017.** An overview of spatial variability of soil microbiological properties using geostatistics. *International Journal of Current Microbiology and Applied Sciences* **6(4)**:1132–1145 DOI 10.20546/ijcmas.2017.604.140.

**Jongman RH, Ter Braak CJF, Van Tongeren OFR. eds. 2007.** *Data analysis in community and landscape ecology*. New York: Cambridge University Press, 321.

**Lauber CL, Strickland MS, Bradford MA, Fierer N. 2008.** The influence of soil properties on the structure of bacterial and fungal communities across land-use types. *Soil Biology and Biochemistry* **40(9)**:2407–2415 DOI 10.1016/j.soilbio.2008.05.021.

**Ming-Huang P, Li Y, Sumner M. eds. 2012.** *Handbook of soil sciences: resource management and environmental impacts*. Second edition. Boca Raton: CRC Press, 818.

**Nannipieri P, Pietramellara G, Renella G. eds. 2014.** *Omics in soil science*. Norfolk: Caister Academic Press, 198.

**RStudio Team. 2012.** *RStudio: integrated development for R*. RStudio, Inc., Boston, MA. *Available at* http://www.rstudio.com/.

**Shange RS, Ankumah RO, Ibekwe AM, Zabawa R, Dowd SE. 2012.** Distinct soil bacterial communities revealed under a diversely managed agroecosystem. *PLOS ONE* **7(7)**:e40338 DOI 10.1371/journal.pone.0040338.

**Treseder KK, Balser TC, Bradford MA, Brodie EL, Dubinsky EA, Eviner VT, Hofmockel KS, Lennon JT, Levine UY, MacGregor BJ, Pett-Ridge J, Waldrop MP. 2011.** Integrating microbial ecology into ecosystem models: challenges and priorities. *Biogeochemistry* **109(1–3)**:7–18 DOI 10.1007/s10533-011-9636-5.

**Umali P, Oliver DP, Forrester S, Chittleborough DJ, Hutson JL, Kookana RS, Ostendorf B. 2012.** The effect of terrain and management on the spatial variability of soil properties in an apple orchard. *Catena* **93**:38–48 DOI 10.1016/j.catena.2012.01.010.

**Wallenius K. 2011.** Microbiological characterisation of soils: Evaluation of some critical steps in data collection and experimental design. D. Phil thesis, University of Helsinki. *Available at* http://ethesis.helsinki.fi/.