# Peer review of "Bacterial and archaeal spatial distribution and its environmental drivers in an extremely haloalkaline soil at the landscape scale"

_PeerJ, doi:10.7717/peerj.6127_

## Round 0.1 · original submission · Major Revisions

I read through the highly constructive comments from two reviewers. I agree with them that there are merits for these data and the associated results obtained. I also agree that there are a lot of room in improving the manuscript quality. For instance, it is suggested that the manuscript could start with hypotheses to lead the readers reading through, and the revealed patterns could be synthesized concisely in discussion and conclusions.

The results are full of nine figures and five tables, and should be refined and presented in a clearer matter. In my opinion, this total could be cut back to five figures and two tables.

Most importantly, the writing quality, both in terms of grammar and English usage, needs improvement in order to flow better. These comments require a substantial effort, especially in rewriting, but if the authors are ready to tackle this task, I recommend a major revision. Of course, this revision should address all the reviewers’ comments carefully.

Reviewer 1 ·

Basic reporting

One of the barriers to smooth reading of the manuscript was incorrect grammar throughout. The authors should ensure they have a fluent English speaker read over and edit their manuscript.

Overall, there were sufficient literature references and background provided to set up the stage of the research.

The outline of the manuscript follows an acceptable format. The authors should ensure that for each figure and table, any abbreviations used are defined. There are several figures/tables where this wasn't done, including the supplemental figures/tables and Table 5. Additionally, several figures have incorrect legends including Figs. 3 and 7. The authors did submit raw sequence data to a public depository.

The authors presented an objective rather than a hypothesis. The results were relevant to this objective, however, presenting a hypothesis could make the paper stronger, more convincing, and provide a framework in which to focus on in the paper.

Experimental design

The authors do present original primary research on an objective that is stated at the end of the introduction. The authors present several research gaps such as a lack of information related to Archaea in hypersaline soil environments and a lack of examination of microbial communities and phsicochemical properties at their specific research site. The manuscript could be improved if the authors could better address why understanding Archaea distribution in hypersaline soils is important.

The experimental design overall seems okay, however, there were some shortcomings. First, I was left unconvinced as to why and how adding the geospatial analysis was beneficial to the study. When looking at a transect, it seems correlation analysis/CCA would be sufficient. Geospatial analysis would be more useful when looking at a study site with 2d collection sites. Second, the description of the methods could be improved in that even steps that are given a reference could be briefly spelled out. Also, the section describing three DNA extraction methods includes no description as to what those methods were or justification for why three methods were used (lines 104-120).

Validity of the findings

The sequence data for the research project were submitted to the NCBI's SRA for public access. The quality of the data I'm sure is relatively robust. An n=13 is not a huge sample size, but sufficient to draw useful results from.

The results of the study need to be refined and presented in a clearer matter. It was difficult to read through. Perhaps ordering the results into the specific pysicochemical parameters would be useful to organize ideas more efficiently. Similarly, the discussion needs to be improved to convey the most important findings from the results. Focusing on maybe just a few patterns would be very helpful in leaving the reader with a clear understanding of the take home message of the research project, and what the next steps are.

Additional comments

The authors have chosen an interesting project to work on, however, the manuscript itself needs a lot of improvement in order to be of publication quality. In general terms, the writing quality, both in terms of grammar and English usage, needs improvement in order to flow better. Also, abbreviations need to be used consistently throughout the manuscript and figures/tables. Regarding specific issues to be addressed, my suggestions have been annotated in the PDF of the manuscript, and stated above.

Annotated reviews are not available for download in order to protect the identity of reviewers who chose to remain anonymous.

·

Basic reporting

There are a number of issues with the basic reporting or the results. A number of the problems are listed below, but the authors need to do a thorough and careful revision to be sure they have located all of the errors in the text. Also the legends on all Figures and Tables need to be checked for accuracy and should include the units in all cases. See specific comments below.
Results
Figure 2: should the last column be water content or water holding capacity (WC)? All the data is reported as WC
Line 219-221 What is the r-value for this negative correlation?
Line 246: reported Chao 1 values are incorrect according to the data in Table 3
Line 247 PD values are also reported incorrectly (range is 4.97-7.37)
Line247 –What do you mean by “taxa diversity” – be more specific. Is this species richness?
Table 3 and lines 245-249: Can you explain why your 4 alpha diversity metrics (OTU richness, Chao, Ace and PD) show different relative relationships among your samples for species richness?
Figure 3: the relative abundance key in your heat map appears to be backwards – red should be the most abundant not the least.
Lines 255-256 – what correlations were used here (Spearman’s?) and please designate the r or rho values so the strength of the correlation is indicated.
Figure 5 Clarify figure so the strength of correlations is understood (r or rho value) and whether correlations are positive or negative.
Line 256 Spirochaetes is positievely correlated with what?
Line 257: The statement that WC is significantly correlated with “most bacterial phyla” does not correspond to the data - only significantly correlated with 5 phyla which is less than half of those listed
Line 257-258: Clarify correlations of Chorobi with soil properties (<50% of those listed)
Line 262 Silt content of TX 010 higher than that of TX008. Check your analyses against your data. There are many statments in the Results section that do not correspond to the data presented in your figures and tables. A more careful analysis of your data is required before this work can be published
Lines 265 -266: this statement is not supported by the heat map in Figure 5A for Chlorobi, WS2 and OD1
Lines 275-279 Please define the sill/nugget ratio and the significance of the high and medium rankings.
Line 281 A pH of 10.2 may be lower than the narrow pH range of 10.3 to 10.6, but it is hard to believe that this difference would make a significant impact on relative abundances of taxa. Are you sure that pH is not conflated with something else?
Line 285 Check the easting values for the low predicted EC. These values do not correspond to figure 7
Lines 299-300 Where is the Shannon ratio of Bacteria:Archaea presented in the results?
Line 304: As stated above, the pH range is very narrow and it is hard to believe it would affect the relative abundance of Thaumarcheota. They also appear correlated with lower EC values
Lines 305-311. Please put labels for the observed values on the kriging plots so that the reader can follow this discussion.
Figures 8 and 9: for Kriging figures please clarify the units of observed values

Experimental design

Specific Comments
Introduction
Line 41: grammar correction. Soil is spatially and temporally the most heterogeneous environment on earth
Lines 44-45: Do you mean the “weather forms the soil” or do you mean to say that – at a global scale, weather determines soil diversity. Please clarify this sentence
Line 53: grammar correction: change to geochemical contribution to microorganism distribution.
Line 55-56 change to, the determination of … and their relationship to
Line 64 change to, can grow
Line 72 change to, make
Lines 74-77 remove italics for cited authors not in parentheses

Methods
The experimental design of this research is good.
Lines 99-100: EC is typically determined from the supernatant not a saturated soil extract, but pH can be determined from a saturated extract. Is this reported correctly?
Line 124: don’t you mean that sequences <290 nt and >530 nt were removed. It doesn’t make sense as written. Same for archaea
Line 134 assignment was determined using
Line 139 change assignations to assignments throughout paper
Line 143 these sequence numbers are extremely low. Please explain

Validity of the findings

Specific Comments

Discussion
Line 324-326 Why was organic C not mentioned in Lines 280-290 of the results where it is stated “other measured factors were more homogeneous – other than pH, EC and WC?
Lines 340-345 As stated earlier, your alpha diversity metrics are inconsistent. Tx008 and Tx012 have low Shannon and Chao 1 values but high observed richness (OTUs). Also there is quite a range of Chao 1 values. I would not call this “similar species richness and diversity in most samples”. You should explore differences between richness and evenness to explain the different patterns.
Line 348-349: Figure 2 shows a correlation of Cenarchaeum with sand and silt but not clay and it is not clear from the heat map whether this correlation is negative or positive.
Line 358-359: Which diversity metric are you using to show that Archaeal diversity correlates with WC, total C and inorganic C? Table S1 cites pH, WC, EC and clay. Your use of multiple analysis tools and the errors in some of the presentation of results as outlined above make it difficult to follow your conclusions. A careful revision is needed to correct errors in this manuscript and label the figures as directed above so that correlations can be more easily followed.
Line 369 – what about Gemmatimonadetes (Figure 3). This phylum appears more abundant or as abundant as Bacteroidetes.
Line 391 -396 Again, you have to point out that your site does not have a pH gradient. 10.3 – 10.6 is not a gradient
Lines 398-414: while it is relevant to explain how these organisms can survive under these extreme conditions, the authors also need to explain their didtribution across the transect since that it is the objective of this manuscript.
Line 415 Which diversity metrics are you using to support this statement. As stated above you need to discuss richness and evenness (OTUs, Chao vs Shannon). In a few sites Archaea diversity was higher for some metrics.
Lines 422-431 – this section should discuss the correlation between the spatially heterogeneous environmental parameters such as EC, WC, and orgC and the respective distribution of bacterial or archael phylotypes

Conclusion
Line 441-443: The statement that bacteria have a more homogeneous distribution along the sampling transect due to patterns for Proteobacteria and Actinobacteria is an unfortunate focus since Chlorobi and Firmicutes are highly variable in relative abundance (Figure3) ranging from 0.01 to 25%. The variability appears high for Firmicutes in Figure 9.

Additional comments

The objective of this study was to examine environmental drivers influencing the spatial variability of microbial populations along a 211 m transect through former Lake Texcoco in Mexico. The transect included significant variability in EC, water content (WC) and org C. Efforts to model the spatial heterogeneity of microbial communities within a limited area are valuable because microbes operate at a pore scale and thus can be influenced by pore-scale variations in soil physicochemical properties. The authors used an interesting combination of statistical tools to correlate the variability in bacterial and archaeal relative abundances with variations in soil properties. They also used Kirging to try and predict the observed variation. While the Methods used in this manuscript are good, the reporting of results is poor because there are a number of statements in the Results section that do not correspond correctly to the data presented in the figures and tables as described under specific comments. In addition, there are errors in the figures as described below. Finally, the authors do not do a good job in the Discussion of synthesizing the patterns revealed by each of the analysis tools, thus it is difficult to follow such comments in the discussion as the statement in lines 358-359 that Archaeal diversity corresponds with water content. The research has merit because the experimental design is good and the data generated appear to reveal some interesting patterns, however a major revision is needed to correct all of the errors, improve the figures and write a more focused discussion to indicate how all the different analyses can be integrated to explain the observed herterogeneity in the distribution of bacterial and archaeal relative abundances.

---

## Round 0.2 · Major Revisions

Please address the comments of the reviewer. As suggested by the reviewer, the discussion still needs considerable improvement. As it is now, it is hard to read the discussion because there is a lack of order and flow. Furthermore, the overall conclusion needs to be more focused so that the reader comes away with the main take-home points of the paper.

The language is still a issue.

The figures could be better by reorganiztion. Some figures are quite messy, such as Figure 3. For instance, three panels in Fig. 1 were shown in a column and the text is too small to be read. Using abbreviations, such as WC, EC, in figures, would be helpful.

There are still statistical problems due to the low number of samples. Overexplanation could be found in CAP analyses (Figures 1, 3) by including non-significant variables and showing very high explanation variations. Operation room is very small due to the low sampling efforts. With only 13 samples, the extrapolations in such a small geographical scale could not be solid (Figures 4, 5). More samples and higher sampling efforts are needed for real geostatistical analyses.

Reviewer 1 ·

Basic reporting

The authors have submitted a "major revision" of a previously submitted manuscript. Overall, the authors have addressed most of the suggestions.

The English has been improved in this revision of the manuscript, however, there were still some errors, some of which have been corrected in the annotated manuscript I am attaching.

The results are relevant to the hypothesis (lines 89-91). However, the term "great spatial variability in physicochemical characteristics" (pH, EC) is misleading because the pH only varies from 10.3 - 10.9. Perhaps this should be discussed in the introduction. Furthermore, microbial community structure should be expanded to include diversity.

Consistent use of abbreviations is suggested. Often the authors go back and forth between water content and WC, or phylogenetic diversity and PD. Please see lines 444, 285-286, and 515.

Some sentences are confusing or out of place. The sentence in lines 481-483 seems out of place and does not fit in with the rest of the paragraph. The sentence in lines 491-492 is confusing and should be reworded. The legends in Fig 1 - 3 are confusing and should be reworded for clarity. For example, Fig 2 A) does not just show significant correlations, it shows all correlations. It is suggested that either the authors only show the phyla/genera that have significant correlations with physicochemical properties, or that it be reworded to, for example, "spearman correlations between a) phyla or b) genera. Significant correlations are displayed as red (negative) or blue (positive) (p<0.05)".

For Fig. 3, the A and B should be A and C, and the C and D should be B and D. Please ensure that all Figures and references to the Figures are correct.

Experimental design

The experimental design is sound. The authors have added more detail and explanations as suggested after first review.

Validity of the findings

For the discussion, rather than starting with the "soil characteristics", which is basically a repeat of the results (lines 380-389), it would be more beneficial to begin with a paragraph presenting the major findings to help guide the reader and provide a focus.

The paragraph starting on line 408 should be more specific as to which soil property affects which soil microbial communities. Also, the term "A lot of" is not very professional. Also in this paragraph, ammonium oxidzation is introduced, without a discussion of how this function relates to this environment, or why it would dominate in areas with lower pH and EC. (Also, saying lower pH is misleading because there really is no pH gradient).

In lines 446 - 452, microbial activity is introduced, however, this is not a focus of the paper.

In lines 453-454, does WC "control" the microbial community? Or is there just a correlation? A

The paragraph in lines 491 - 499 is out of place in the discussion, and should be moved to the results. Also, I think the purpose of this paragraph is to explain why there are low sequence counts, however, this needs to be more explicit. Also, lines 496-498 do not support why there would be low sequence counts in the extreme hi saline soil of this study, since in the referenced study there are high sequence counts.

Additional comments

Overall, the experimental design and the results presented seem robust, however, the discussion of them still needs great improvement. As it is now, it is hard to read through the discussion because there is
a lack of order and flow. Furthermore, the overall conclusion needs to be more focused so that the reader comes away with the main take-home points of the paper.

Please see some minor comments in the attached pdf as well.

Annotated reviews are not available for download in order to protect the identity of reviewers who chose to remain anonymous.

---

## Round 0.3 · Minor Revisions

Per the discussion among editors, the low sample number or sequence depth would not be a key issue for this manuscript. In the discussion section, however, please provide caveats to the readers regarding the low sampling number for geostatistical analyses.

Please carefully reply to the comments on statistical analyses to make sure that statistical aspect is performed to the technical standard required for publication.

(1) For the clarity in the statistical analyses, please mention clearly how “canonical analysis of principal coordinates” was performed and how the explanatory variables were selected for the final model. Also, please mark down the significance for the selected explanatory variables in CAP figures.
(2) In CAP figures, readers could not assign the variables to arrows clearly. For example, which arrows should for Halobacteriales Halorrhabdus and Haloterrigna” in Figure 1B? The same for environmental variables.
(3) Did you miss “silt content” for correlation analyses? From CAP, the silt content was very important.
(4) The proportions of OTU/sequences were extremely high (Table 1 and 2). For example, you found 10533 OTUs out of 11419 Archaeal sequences at the 97% level. There were 4439 OTUs out of 5121 bacterial sequences. Could you explain this? Please provide detailed results of sequence analyses to support this. Please discuss this phenomenon in the discussion by referring to previous literature or public data if possible.
(5) In Table 1 and 2, if the OTUs were normalized to lowest sequences, how could the OTU numbers be larger than the lowest values of sequences (that is 231 and 160 for bacteria and archaea, respectively). Please explain it.
(6) Provide “Coverage values” for archaeal data (Table 1).


There are still other issues as below.

Line 218. If the abbreviation LOO was not going to be used in following text, remove it.
Line 305. Phylogenetic diversity. Explain how PD was calculated in material and methods section.
Line 276. Once the abbreviation PD is used, PD could be used in following text. No need to mention “Phylogenetic diversity” again (line 305).
Cite proper references for Chao1 and AEC. Mention these metrics in material and methods section.
Figure 1. The location of :Water content (WC)”.
Figure 1. Full line? It should be solid line. Explain the line types in legends.
Figure 1, 3. Species scores? Are you considering “species”?

---

## Round 0.4 · accepted · Accept

Congratulations. Your manuscript is now accepted after revision.

#